

# Overcoming the IOTLB wall for multi-100-Gbps Linux-based networking

Alireza Farshin[1], Luigi Rizzo[2], Khaled Elmeleegy[2] and Dejan Kostić[1]

[1] School of Electrical Engineering and Computer Science, KTH Royal Institute of Technology, Kista, Sweden
[2] Google, Mountain View, California, United States of America

## ABSTRACT

This article explores opportunities to mitigate the performance impact of IOMMU on high-speed network traffic, as used in the Linux kernel. We first characterize IOTLB behavior and its effects on recent Intel Xeon Scalable & AMD EPYC processors at 200 Gbps, by analyzing the impact of different factors contributing to IOTLB misses and causing throughput drop (up to 20% compared to the no-IOMMU case in our experiments). Secondly, we discuss and analyze possible mitigations, including proposals and evaluation of a practical hugepage-aware memory allocator for the network device drivers to employ hugepage IOTLB entries in the Linux kernel. Our evaluation shows that using hugepage-backed buffers can completely recover the throughput drop introduced by IOMMU. Moreover, we formulate a set of guidelines that enable network developers to tune their systems to avoid the "IOTLB wall", *i.e.*, the point where excessive IOTLB misses cause throughput drop. Our takeaways signify the importance of having a call to arms to rethink Linux-based I/O management at higher data rates.

## INTRODUCTION

The introduction of multi-hundred-gigabit network interfaces (100/200/400 Gbps) has motivated many researchers to study networked systems & their underlying hardware to understand the challenges of achieving suitable performance at higher data rates. These studies have explored various topics, such as Linux kernel network stack (*Cai et al., 2021*), Peripheral Component Interconnect Express (PCIe) (*Neugebauer et al., 2018*), Remote Direct Memory Access (RDMA) (*Kalia, Kaminsky & Andersen, 2016*), Data Direct I/O (DDIO) & Last Level Cache (LLC) (*Farshin et al., 2019, 2020*), and smart Network Interface Cards (NIC) (*Katsikas et al., 2021*; *Liu et al., 2019*). Most studies focused on improving performance and paid less attention to other aspects, such as security and privacy, which are critical real-world requirements. Consequently, these studies often disable many software & hardware features to maximize the performance of networking applications. This article examines one of these features, specifically the I/O Memory Management Unit (IOMMU), which facilitates input-output (I/O) virtualization and provides I/O security, to quantify its performance overheads at multi-hundred-gigabit rates.

Corresponding author
Alireza Farshin, farshin@kth.se

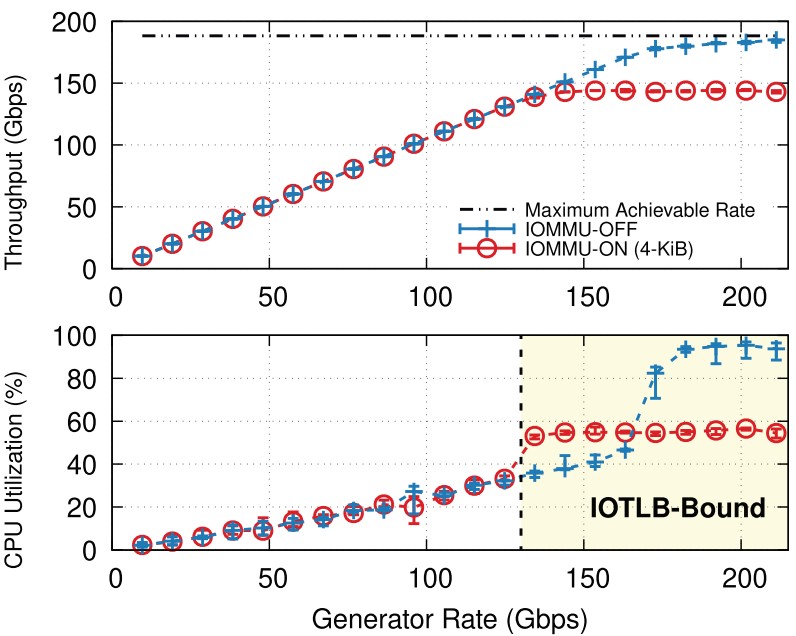

**Figure 1 TCP throughput and receiver CPU utilization with and without IOMMU on a receiver (200-Gbps link, MTU = 1,500).** Enabling IOMMU on a receiver puts a hard limit on the TCP throughput. Note that the receiver's CPU utilization (lower graph) is well below saturation.

**Why IOMMU?** IOMMU makes it possible to use virtual addresses rather than physical addresses when performing I/O *via* Direct Memory Access (DMA), thereby creating an abstraction layer between the I/O devices and the processor. IOMMU provides hardware, similar to the Memory Management Unit (MMU) and Translation Lookaside Buffer (TLB), to translate I/O Virtual Addresses (IOVA) to their respective physical addresses (PA). This can be used to restrict DMAs from I/O devices to only limited regions of the processor's memory address space (*i.e.*, the pages mapped to the IOMMU hardware), hence (*i*) providing a higher degree of protection and (*ii*) easing virtualization & backward compatibility. This comes at the cost of performance: Figure 1 shows how the throughput and Central Processing Unit (CPU) utilization on a Transmission Control Protocol (TCP) receiver (*i.e.*, iPerf server) vary with offered rate. Without IOMMU, the receiver almost reaches the line rate before becoming CPU bound. Enabling IOMMU on the receiver caps the throughput causing a significant throughput drop, not due to CPU overload (utilization remains around 60% in this experiment), but due to reasons explained in "IOMMU Performance Impact on DMA Operations".

**What is new?** Previously evaluations of the IOMMU performance mainly focus on CPU overheads, such as (*i*) IOVA (de)allocation (*Malka et al., 2015a*; *Malka, Amit & Tsafrir, 2015b*; *Markuze et al., 2018*) and (*ii*) I/O TLB (IOTLB) invalidation (*Markuze, Morrison & Tsafrir, 2016*; *Peleg et al., 2015*). A common optimization is to make mapping entries more long-lived, thereby making the most out of the IOTLB and addressing security concerns. In particular, DAMN (*Markuze et al., 2018*) presented a DMA-aware memory allocator that used permanently mapped buffers for packet buffers to re-use pages for DMA-ing. DAMN

primarily focused on 4-KiB pages and measured IOMMU performance only for 9000-B jumbo frames. Our work looks at a different aspect of the problem, *i.e.*, data path impact of IOMMU and IOTLB misses, which does not involve the CPU overheads. We characterize such an impact in a variety of scenarios affecting the IOMMU translations and IOTLB locality different data rates, Maximum Transmission Unit (MTU) sizes, and CPU types and propose mitigations to recover the throughput drop introduced by the use of IOMMU. Our proposed solution explores I/O security *vs.* performance trade-offs *and* implementation challenges in using IOMMU mappings with larger granularity (*e.g.*, 2-MiB "hugepages") in the Linux kernel, at much higher data rates than explored in previous works. Note that our approach is orthogonal to prior works discussing the challenges of allocating hugepages *and* benefits of using them to reduce TLB (*i.e.*, MMU translation cache) misses (*Kerrisk, 2013*; *Panwar, Prasad & Gopinath, 2018*).

**Why now?** The root cause of the throughput drop, described in "IOMMU Performance Impact on DMA Operations", becomes more and more serious as NIC link speeds grow to 200 Gbps and above. Previous works that addressed similar problems (*Amit, Ben-Yehuda & Yassour, 2012*; *Ben-Yehuda et al., 2007*) worked with 10 times lower link speeds and more limited hardware (*e.g.*, Intel Xeon Paxville & Nehalem and AMD Opteron), thus those measurements and findings are not necessarily applicable to current systems. Our measurements on a variety of recent processors and NICs are thus essential to better understand the impact of the problem today.

**Contributions.** In this article, we:

- Provide a thorough characterization of the performance impact of IOMMU at high network speeds, and its dependency on multiple parameters (*e.g.*, offered load, MTU, packet drops, and network features offloads) and different hardware (*e.g.*, processors & NICs);
- Carefully investigate the challenges of employing hugepages in Linux network device drivers, to find a sweet spot between memory stranding, page locality, and CPU overhead;
- Propose a hugepage-aware memory allocator (called HPA) for device drivers and show its potential benefits;
- Demonstrate the effectiveness of employing hugepages to mitigate IOTLB overheads at 200 Gbps. Our memory allocator eliminates the up-to-20% throughput drop caused by the default configuration of the IOMMU.

**Takeaways.** The main takeaways of our studies[1] are:

① IOTLB misses put a hard limit on achievable throughput as described in section "IOMMU Performance Impact on DMA Operations". Once we reach this limit, which we call the "IOTLB wall", IOTLB misses have a sudden increase.

② Reducing the number of IOTLB misses can significantly shift the bit rate at which the system starts experiencing throughput drops ("Data Rate and IOTLB Wall").

[1] The source code is available at github aliireza/iommu-bench.

③ Using Large Receive Offload (LRO) effectively reduces IOTLB overheads for sub-4-KiB buffers (section "Large Receive Offload").

④ Enabling TCP Segmentation Offload (TSO) can mitigate IOTLB overheads for MTU sizes < 3,000 B (section "TX & TCP Segmentation Offload").

⑤ Packet drops and TCP re-transmission shuffle buffers, increasing IOTLB misses (section "Other Workloads").

⑥ Network device driver implementation affects the number of IOTLB misses (section "Different NICs"). Additionally, some network drivers are unable to recycle their buffers, causing continual page allocations and reducing locality, which can increase IOTLB misses (section "Proposed Solution" & "Run-Time Allocation Overhead").

⑦ If the driver is not leaky (*i.e.*, it can operate with a pre-allocated memory pool), the memory pool should perform locality-aware recycling to minimize IOTLB misses due to "buffer shuffling" (section "Buffer Shuffling").

## BACKGROUND

IOMMU provides hardware support for I/O virtualization, allowing DMA operations to use virtual addresses rather than physical addresses. IOMMU hardware is present in most recent processors (*e.g.*, Intel VT-d (*Intel, 2021*), AMD-Vi (*AMD, 2021*), and ARM SMMU (*ARM, 2021*)), and typically performs two main tasks: (*i*) DMA remapping and (*ii*) interrupt remapping.

**DMA remapping** maps virtual addresses in DMA operations to physical addresses in the processor's memory address space. Similar to MMU, IOMMU uses a multi-level page table to keep track of the IOVA-to-PA mappings at different page-size granularity (*e.g.*, 4-KiB, 2-MiB, and 1-GiB pages). The hardware also implements a cache (*aka* IOTLB) of page table entries to speed up translations. IOTLB features vary across different processors, with little if any public documentation. The available performance counters and public data sheets suggest that Intel processors use a single IOTLB for all levels of mapping (*e.g.*, first-level, second-level, nested, and pass-through mappings) (*Intel, 2021*), whereas AMD processors use two distinct IOTLBs for caching Page Directory Entry (PDE) and Page Table Entry (PTE) (*AMD, 2021*; *Kegel et al., 2016*). Additionally, *Emmerich et al. (2019)* and *Neugebauer et al. (2018)* have speculated (based on their experiments) that the number of IOTLB entries for some Intel processors is 64 and the cost of an IOTLB miss & its subsequent page walk is around 330 ns. Furthermore, some PCIe devices support Address Translation Service (ATS) (*Krause, Hummel & Wooten, 2006*; *PCI-SIG, 2009*) that enables devices to cache the address translation in a local cache to minimize latency and provide a scalable distributed caching solution for IOMMU. Based on our discussions with NVIDIA/Mellanox support, since Intel IceLake processors only caching of 1-GiB IOTLB entries is supported *via* ATS (*i.e.*, this caching is not helpful for 4-KiB entries).

**Interrupt remapping** enables I/O devices to trigger interrupts by performing DMAs to a dedicated memory range (*Bugnion, Nieh & Tsafrir, 2017*; *Hafeez & Patov, 2017*); thus, VMs can receive interrupts from I/O devices.

**Use cases.** IOMMU is used for inter-guest and intra-guest protections. The former provides inter-VM protection in virtualized settings. Whereas the latter is related to DMA attacks. For instance, when IOMMU is not enabled, NICs with malicious or buggy firmware could extract information. This article focuses on non-virtualized settings. Besides providing isolation and protection against DMA attacks, IOMMU can be used to (*i*) support 32-bit PCIe devices on x86-64 systems, (*ii*) hide physical memory fragmentation, and (*iii*) perform scatter-gather I/O operations.

**Monitoring capabilities.** Intel VT-d and AMD-Vi provide performance counter registers to measure different IOMMU-related metrics, such as the number of hits & misses in the IOTLB and in various levels of the I/O page table.

## IOMMU performance impact on DMA operations

**Basic model.** I/O subsystems have an upper bound on the number ($N$) of outstanding DMA transactions. This implicitly creates a throughput bottleneck $B \propto N/(T_{MEM} + T_{IOMMU})$, where $T_{MEM}$ is the time to complete a transaction without IOMMU, and $T_{IOMMU}$ is the time to perform related address translations. In ideal conditions, $B$ exceeds the desired data rate and the capacity of other buses (*e.g.*, NIC & PCIe) in the data path; but exceedingly high translation times (*e.g.*, due to excessive IOTLB misses) can reduce $B$, making it the bottleneck.

**Dependencies.** $T_{IOMMU}$ is negligible if a translation can be served by the IOTLB; otherwise, it depends linearly on the probability and number of IOTLB misses per translation. Misses in turn are heavily affected by (*i*) the size and management policy of the IOTLB (unfortunately, these are unmodifiable hardware features) and (*ii*) by the memory request patterns (and implicitly the IOTLB request patterns), which we study and optimize in this article.

**IOTLB miss metrics.** The absolute number of IOTLB requests and misses can be determined using dedicated performance counters on the processor. Depending on the data direction and system configuration, each block of data may require one or more translations. Additionally, many factors (*e.g.*, packet & MTU size, buffer organization, packet rate, and drop rate) could substantially affect the number of IOTLB misses. As a consequence, neither the absolute value nor the percentage of IOTLB misses is especially meaningful to assess their effect or compare system behavior with different configurations. Instead, we found that a metric of IOTLB misses per MiB is especially convenient, as it reflects the number of IOTLB misses per unit of data after dividing it by the throughput, making it possible to compare the impact of IOTLB misses across different system configurations (*e.g.*, different MTUs, packet rates, and buffer organizations).

**Buffer shuffling and positive feedback loops.** Common buffer management strategies try to reduce IOTLB misses by improving buffer address locality. However, in the presence of packet drops and re-transmissions, data buffers are not recycled immediately, which shuffles buffers and reduces their locality, thereby causing more IOTLB misses, more (receiver-based) packet drops, and more shuffling in a positive feedback loop. Consequently, the system may enter into a poor performance regime from which it is difficult to escape, see the "Other Workloads" and "Buffer Shuffling" sections. The effect of

buffer shuffling may be amplified in scenarios with more than a single buffer pool (*e.g.*, multi-queue network device drivers with a separate pool per queue).

## Networking data path in the Linux kernel

We briefly summarize the data path in the Linux kernel for both receive (RX) & transmit (TX) paths (*Cai et al., 2021*; *Markuze et al., 2018*).

### RX path

To receive packets from the wire, the network driver initially populates the RX queues of the NIC with pointers to receive buffers allocated from host memory. The size of receive buffers and their number per queue (*aka* RX descriptors) are configurable at run-time. Upon incoming packets, the NIC issues DMA transfers over the bus (typically PCIe) to move incoming data from the NIC's internal buffers to the receive buffers in host memory (or cache (*Farshin et al., 2020*)). When one or more DMA transfers are complete, the NIC issues an interrupt, and the CPU core running an interrupt driver attaches newly filled receive buffers to a packet descriptor (`sk_buff`) and passes them to the kernel for further protocol processing. The RX queue is then replenished with new receive buffers. Eventually, applications will issue a `recvfrom()`, or an equivalent system call, to consume received buffers. In non-zero-copy cases, issuing a receive system call will copy the content of in-kernel receive buffers to userspace buffers, making it possible to immediately return the backing memory to the kernel. However, using a receive zero-copy mechanism could delay returning the memory to the kernel, as the application may hold buffers longer. To the best of our knowledge, there are only a few experimental techniques (with many limitations) to perform receive-side zero-copying within the Linux kernel. These techniques require applications to return the buffers within a reasonable time; otherwise, other optimizations to reduce memory mapping costs would break. Therefore, most zero-copying applications rely on userspace network stacks and kernel-bypass frameworks, which target a different domain than our work. "Frequently Asked Questions" elaborates on the importance of having high-performance Linux-based solutions alongside kernel-bypass frameworks.

**Receive buffer management.** NIC device drivers use a custom or general-purpose RX buffer pool (*e.g.*, the Page Pool Application Programming Interface (API)[2] (*The kernel development community, 2021*)) to speed up the allocation and deallocation of receive buffers. The RX buffer pool generally implements a per-CPU-core cache of receive buffers, with optimizations to re-use and *recycle* the receive buffers once they are consumed by applications. Buffer recycling reduces calls to the kernel's page allocator, which may improve memory access locality and, in turn, reduce MMU and IOMMU translation costs.

Each RX buffer pool is used concurrently by multiple CPU cores (*e.g.*, the core serving interrupts requests new buffers, while cores running applications release buffers after completing `recvfrom()` system calls). Thus, the RX buffer pool may choose to trade recycling efficiency for reduced locking contention. Additionally, applications may be slow in consuming data, consequently depleting the cache and forcing new allocations.

[2] See Appendix C for implementation details.

Moreover, buffers are often delivered to different applications, hence released in a different order, which over time may cause buffer shuffling and reduces address locality.

**Receive buffer lifetime.** If a non-zero-copy application is *not* starved of CPU or poorly designed, a receive buffer's lifetime will be reasonably short, making it possible to recycle it rapidly. However, packet drops and TCP re-transmissions would cause the buffers to be held for one or more RTTs, delaying the buffer's recycling. As mentioned in "IOMMU Performance Impact on DMA Operations", buffer shuffling can increase the number of IOTLB misses. "Other Workloads" further evaluates the impact of packet drops on IOTLB misses. This work does not focus on kernel-based applications that hold kernel buffers for unreasonably long times, as they have much worse throughput bottlenecks than IOMMU/IOTLB.

### TX path

Transmissions are initiated by applications calling `sendmsg()` or equivalent system calls, which allocate in-kernel transmit buffers to copy user-supplied data, attach them to `sk_buffs`, and pass them to the network stack; some implementations may use zero-copy mechanisms, see "Frequently Asked Questions". Eventually, the addresses of the data buffers will be passed to the NIC's TX queue so that the NIC will be able to issue DMA transfers and perform the transmission. When the transmission is complete, the NIC issues an interrupt and the device driver can return the buffer memory to the kernel.

**Transmit buffer management.** Similar to the RX path, the TX path may use caches to allocate/release buffers. The constraints are slightly different, as in this case there may be multiple CPU cores allocating buffers, and only one core releasing them.

### Operation when IOMMU is enabled

When IOMMU is enabled, the driver must create IOMMU mappings for receive buffers & descriptors, and pass IOVAs instead of physical addresses to the NIC. The granularity of these mappings is constrained by the granularity of physical pages backing these memory buffers. Currently, DMA mapping is mainly performed at the 4-KiB page granularity. For an IOMMU address space of 48 bits, one address translation requires accessing four entries in the IOMMU page table[3]. When a buffer is not used for DMA anymore, its IOMMU mapping is no longer needed. Depending on the protection *vs.* performance trade-off policy, the operating system may decide to remove the mappings right away (*i.e.*, a very expensive operation, since it involves flushing the IOTLBs); do the unmappings in batches at a later time; or keep the mappings alive in case the buffers are recycled. Note that deferring the unmappings slightly relaxes the security guarantees provided by IOMMU, but reduces its overheads (*Peleg et al., 2015*).

## IOMMU PERFORMANCE CHARACTERIZATION

As reported in previous works (*Ben-Yehuda et al., 2007*; *Markuze et al., 2018*; *Peleg et al., 2015*), enabling/using IOMMU typically causes a drop in network bandwidth, due to factors such as (*i*) IOVA (de)allocation, (*ii*) IOTLB invalidation, and (*iii*) IOTLB misses. Previous works have mainly focused on the first two factors, which typically show up as

[3] A total of 48 bits address space with 8-B entries and 4096-B page-table size requires four page tables (each having 512 entries) (*Malka, 2015*; *Stack Overflow, 2011*).

additional CPU load. This section studies the correlation between the IOTLB misses and the throughput drop, which is independent of CPU load.

Following the description in the section "IOMMU Performance Impact on DMA Operations", IOMMU translation costs do not affect throughput up to the point where serving IOTLB misses takes too long, thus creating a throughput bottleneck. The IOTLB antagonization threshold, *i.e.*, IOTLB wall, depends in part on hardware features we cannot modify: the number and management of IOTLB entries, the maximum number of outstanding translation requests (*aka* I/O page walkers), and the completion time of an IOMMU translation reading entries from main memory. A second dependency is on the request pattern, which is partly under our control. Since IOTLB is a cache of the IOMMU page tables, we can reduce misses by managing buffers in a way that improves address locality. As an example, placing sub-4-KiB buffers belonging to the same 4-KiB page in consecutive RX descriptors generally saves at least one IOTLB miss for descriptors after the first one. Reducing the total number of buffers, making full use of buffer space, batching accesses to descriptors, all may contribute to reducing the number of IOTLB misses.

This section characterizes IOTLB behavior in a variety of scenarios, to better understand the correlation between the number of IOTLB misses and the throughput drop. More specifically, we study the factors that could affect the translation working set, consequently affecting IOTLB locality and throughput. We mainly rely on iPerf, a network performance measurement tool, to run microbenchmarks and characterize the performance of IOTLB. We use iPerf due to its efficiency in saturating a 200-Gbps link, whereas real-world applications (*e.g.*, Memcached) typically cannot reach line rate due to other throughput bottlenecks. The section "Other Workloads" shows that our iPerf analysis would still be applicable to less I/O intensive applications. Most of our experiments use Intel Xeon IceLake processors with NVIDIA/Mellanox NICs, but "Different Processors" and "Different NICs" examines other Intel & AMD EPYC processors and a 100-Gbps Intel E810 NIC, respectively.

**Testbed.** Our testbed uses two workstations directly connected *via* a short 200-Gbps cable (or 100-Gbps one in a few experiments); the RTT between these two workstations is negligible. One workstation acts as a traffic source/sink, without IOMMU enabled for maximum performance. The other one acts as the Device Under Test (DUT) and runs different microbenchmarks with & without the IOMMU enabled; in the former case, we use the default setting in Linux for IOMMU (*e.g.*, `intel_iommu=on` that uses deferred IOTLB invalidation (*Peleg et al., 2015*)). Table 1 shows the different processors used for the DUT. In most experiments, the DUT is equipped with an Intel Xeon Gold 6346 @ 3.1 GHz[4] (*i.e.*, IceLake) and a 200-Gbps NVIDIA/Mellanox PCIe-4.0 ConnectX-6[5] ; see Appendix A for more info. The DUT runs Ubuntu-20.04.3 (Linux kernel 5.15). Unless stated otherwise, we set the core frequency to the processor's maximal frequency (*i.e.*, 3.6 GHz) and the uncore frequency to its maximum (*i.e.*, 2.4 GHz), for best memory/ cache access latency (*Gough, Steiner & Saunders, 2015*; *Sundriyal et al., 2018*). To read IOMMU-related performance counters, we use the Performance Counter Monitor (PCM) tool (*i.e.*, `pcm-iio`) on Intel and the Linux `perf` tool on AMD processors. IOTLB hit & miss values and their rates are derived from performance counters sampled once per

---

[4] Nominal frequency.

[5] We use the main slot of a Socket Direct PCIe 4.0 adapter.

**Table 1 Processors used for DUT.** The entry marked with a ⋆ is our primary testbed.

| Generation | Properties | | | |
| --- | --- | --- | --- | --- |
| | Model | Physical cores | Freq. (GHz) | LLC (MB) |
| Intel Xeon Skylake | Gold 6140 | 18 | 2.3 | 24.75 |
| Intel Xeon CascadeLake | Gold 6246R | 16 | 3.4 | 35.75 |
| Intel Xeon IceLake ⋆ | Gold 6346 | 16 | 3.1 | 36 |
| AMD EPYC (3rd gen. Milan) | 74F3 | 24 | 3.2 | 256 |

second. We run our experiments automatically *via* the Network Performance Framework (NPF) tool (*Barbette, 2021*) to improve the reproducibility of our results, and we mainly report the median of five 60-s runs, with min/max error bars (though in many experiments the range is small and almost invisible).

**200-Gbps considerations.** We used the following options to operate iPerf at 200 Gbps without becoming CPU-bound, so our measurements emphasize IOTLB effects: (*i*) enable hyper-threading to increase available CPU cycles (our processors only have 16–24 physical cores each); (*ii*) run all interrupt and application threads on Socket 0, the socket local to the NIC (as the cost of cross-socket memory access exceeds the benefits of using the additional cores on Socket 1); (*iii*) configure the NIC with one TX/RX queue pair and interrupt for each logical core on Socket 0; (*iv*) enable TSO to reduce transmit CPU load, enable Generic Receive Offload (GRO), and enable/disable LRO depending on the configuration, see "Large Receive Offload"; (*v*) enable interrupt moderation mechanisms as appropriate for the platform[6].

**iPerf configurations.** We use iPerf-2.0.14 (*iPerf, 2021*) in TCP mode[7], which uses multiple threads to transfers unidirectional traffic over one or more TCP sockets between one sender and one receiver with configurable packet rate. In our experiments, we tune the number of TCP connections and message blocks to maximize iPerf throughput when IOMMU is disabled; more specifically, we use 384 TCP connections and 128-KiB blocks for read/write system calls (of course, TCP segments these blocks according to window bounds), and report the transport level throughput (*i.e.*, goodput). Note that iPerf messages (in TCP mode) are split into smaller segments based on the Maximum Segment Size (MSS) that is dependent on the MTU size.

**Parameters.** We explore the impact of the following parameters on experiments: (*i*) iommu configuration (off, on, mapping size); (*ii*) offered rate, $R_{IN}$, enforced by rate limiting the sender, which affects both CPU load and IOMMU translation rates; (*iii*) *MTU*, which affects the receive buffer size and IOMMU translation rate for a given offered rate, (*iv*) platform, to study the behavior of different CPUs and NICs. It is worth mentioning that we cannot set packet size for iPerf when operating in TCP mode. Instead, we vary the MTU size, which resembles modifying packet size since it affects the size of transmitted messages. Regarding the impact of packet size, as long as the NIC driver uses the same buffer sizes to receive different packets (*e.g.*, when MTU is not changed), the IOTLB misses

---

[6] We observed that the AMD EPYC 74F3 processor had a hard time dealing with a large number of interrupts per second, which occurs at high packet rates and with a large number of queues. We mitigate the problem by setting the sysctl parameter `napi_defer_hard_irqs=1` that was introduced in the Linux kernel specifically to address that issue.

[7] iPerf uses standard Linux system calls, and in User Datagram Protocol (UDP) mode, it cannot saturate a 200-Gbps link due to high per packet costs caused by per-packet system calls and fewer batching opportunities. Appendix B investigates the IOTLB overheads on both throughput & latency with UDP traffic using DPDK-based FastClick (*Barbette, Soldani & Mathy, 2015*; *Farshin et al., 2021*).

are expected to rise for smaller packet sizes due to a larger number of received packets per second. Appendix B shows the impact of IOMMU for different packet sizes when forwarding forwarding UDP traffic with a DPDK-based application.

MTU experiments use multiples of 1,500 between 1,500 and 9,000 (both commonly used values), plus MTU = 3,690 that is the largest value allowed with 4-KiB buffers for our NIC. We have verified that our testbed can get within 2–3% of line rate at those MTUs[8].

The maximum achievable data and packet rate depend on the MTU. On the wire, each packet occupies the link for $WIRE\_SIZE = (MTU + 38)$ bytes (*i.e.*, 14-B Ethernet header, 4-B CRC, 20-B preamble & inter-packet gap) limiting the maximum packet rate to $link\_rate/(8 \times WIRE\_SIZE)$. Each packet carries a payload of $MSS = (MTU - 52)$ bytes (20-B IP header, 20-B TCP header, 12-B TCP options in IPv4), thus the maximum achievable transport-level throughput ("application line rate" for brevity) is a fraction $MSS/WIRE\_SIZE$ of line rate (*e.g.*, 0.9418 for MTU = 1,500, 0.9900 for MTU = 9,000).

**Metrics of interest** are averaged across the entire duration of the experiment. They include: (*i*) throughput (TP), which deviates from the offered rate in the presence of bottlenecks; (*ii*) throughput drop percentage, $100 \times (TP_{NO\_IOMMU} - TP_{IOMMU})/TP_{NO\_IOMMU}$, which details the relative loss introduced by IOMMU; and (*iii*) *IOTLB misses per MiB*, which indicates how often we incur IOTLB misses. This unusual unit *i.e.*, the iPerf goodput/ number of IOTLB misses, makes it easy to compare the number of misses across different experiments, as discussed later.

## Data rate and IOTLB wall

Our first set of experiments explores the receiver's behavior on IceLake (with 32 logical cores and a 200-Gbps NIC), with MTU = 1,500 B and 1,024 RX descriptors.

Figure 1 shows the average throughput and CPU utilization at different offered traffic rates. Without IOMMU throughput matches the offered rate up to ∼160 Gbps when occasional drops occur, and then CPU load becomes super-linear because of the extra cost to process re-transmissions and out-of-order traffic. At CPU saturation, the system achieves ∼185 Gbps, not far from the theoretical maximum of 188 Gbps. With IOMMU ON, linearity breaks and drops start to appear much earlier, ∼130 Gbps, and throughput eventually tops ∼150 Gbps with CPU utilization never going above 60%.

To explain the throughput drop, which is clearly not due to a CPU bottleneck, we measured, and show in Fig. 2, the throughput drop percentage and the IOTLB misses/MiB at different offered rates.

**IOTLB misses discussion.** In this experiment with MTU = 1,500 & MSS = 1,448, 1 MiB requires $2^{20}/1,448 \approx 725$ RX buffers, each one using half of a 4-KiB page. In normal conditions, it is very likely that contiguous RX descriptors use contiguous half pages; accessing the first half page will cause an IOTLB miss on the last-level IOMMU entry, while the second half page will be served from the IOTLB. This alone causes ∼420 misses/MiB[9]. The remaining 80–100 misses/MiB can be explained by misses on the next-level IOMMU

[8] A very fine-grained sweep of MTU values showed some significant throughput drop with MTU below 1,000, or MTU above 1,500 and below 3,000; see Appendix A. For other values, the testbed behaves as expected, so there is no need to run subsequent experiments with a fine granularity.

[9] One might expect to see $725/2 \approx 362$ misses at most; however, it is important to notice that the network device driver might occasionally or continually allocate new pages due to a slow buffer recycling process, which will further increase the number of misses. Additionally, we use multiple queues, each with a separate buffer pool, which also affects the IOTLB locality.

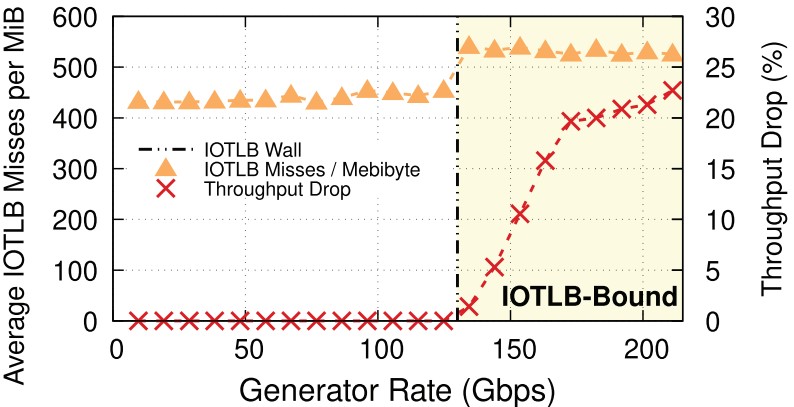

**Figure 2 Throughput drop and IOTLB misses per MiB *vs.* offered rate, MTU = 1,500.** Misses have a sudden jump when the receiver starts dropping packets, because drops create buffer shuffling and in turn reduce address locality.

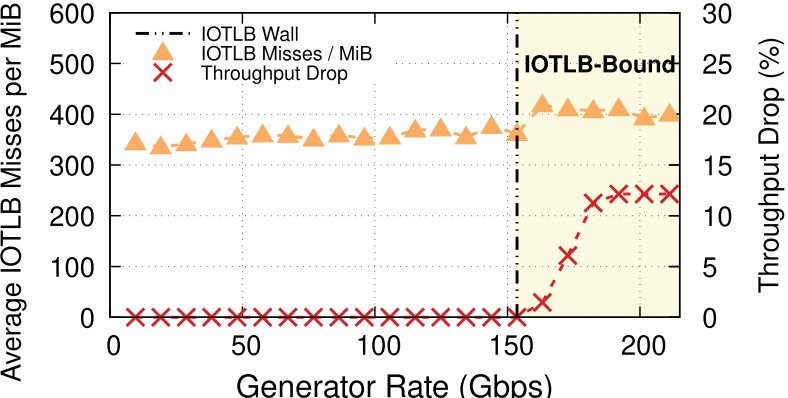

**Figure 3 Throughput drop and IOTLB misses per MiB *vs.* offered rate for MTU = 3,690.** The modest reduction in misses compared to Fig. 2 causes a significant increase in the rate where drops start to appear.

entries, and the interleaving of traffic on a large number of queues, which breaks the regular pattern and causes additional misses also on the next IOMMU level.

When packets start being dropped (at a traffic rate of around 130 Gbps), the misses/MiB suddenly increase. We theorize that the packet drops cause shuffling of received buffers, which in turn reduces locality and thus creates more misses per packet.

To prove our hypothesis, we ran another experiment with 4-KiB buffers (MTU = 3,690). In this case, we expect a last level IOTLB miss on every buffer, even without shuffling, but there are $2^{20}/3,638 \approx 288$ RX buffers per MiB, *i.e.*, many fewer RX buffers than before. Figure 3 provides evidence for our theory, and also shows that the throughput drop occurs at a higher traffic rate (*i.e.*, ~150 Gbps) because of the lower number of misses per MiB. Note how the step in misses per MiB between the two regimes still exists, but it is smaller. The reason for this step is as follows: while shuffling within the same 4-KiB block causes no additional IOTLB miss, there is a large number of 4-KiB blocks in each RX buffer pool (almost double that of the number for the case of MTU = 1,500 because of using 4-KiB

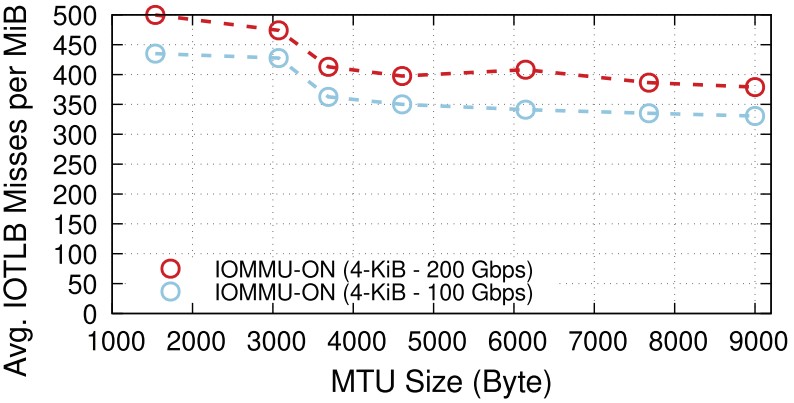

**Figure 4 Receiver side average IOTLB misses per MiB for different MTU sizes, in underloaded (100-Gbps) and overloaded (200-Gbps) conditions.**

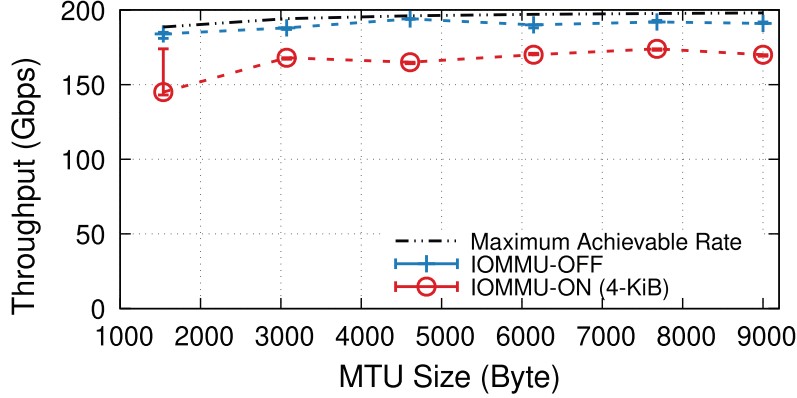

**Figure 5 Throughput with/without IOMMU for different MTU sizes, in overloaded (200-Gbps) condition.**

buffers instead of 2-KiB ones), and shuffling between these blocks does cause extra IOTLB misses.

**Main takeaway.** Comparing Figs. 2 and 3 gives an important piece of information: a modest reduction in IOTLB misses per MiB (e.g., from ∼420 to ∼360 as in our experiments) can significantly shift the data rate at which we start experiencing drops, which will be useful when designing our mitigations.

### Effect of MTU size

We now explore the impact of different MTUs on throughput in two different regimes: underloaded (100-Gbps offered load), and overloaded (200-Gbps offered load). Figure 4 shows the IOTLB miss per MiB for both regimes. The misses-per-MiB follow our expectations, with slightly higher values in the overloaded case, matching the steps we see around the IOTLB wall in Figs. 2 and 3.

When underloaded, with and without IOMMU, throughput matches the offered load, whereas overloading the system can cause additional drops. Figure 5 compares the throughput in the overloaded condition with and without IOMMU, where we see up to

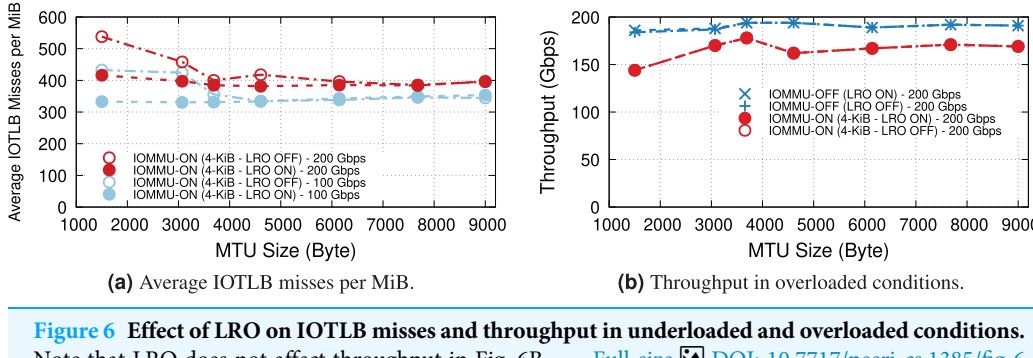

**(a)** Average IOTLB misses per MiB.

**(b)** Throughput in overloaded conditions.

**Figure 6** **Effect of LRO on IOTLB misses and throughput in underloaded and overloaded conditions.**
Note that LRO does not affect throughput in Fig. 6B.

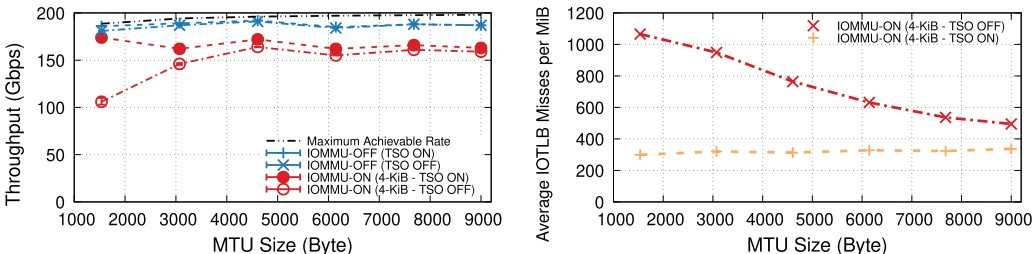

**(a)** Throughput at different MTUs with various combinations of **(b)** Average IOTLB misses per MiB with and without TSO.
IOMMU and TSO on the sender side.

**Figure 7** **Using TSO reduces the IOTLB overheads on the TX path due to fewer IOTLB translations.**

20% drop introduced by IOMMU. It is worth highlighting that the large error bar for the data point representing MTU = 1,500 shows that various system parameters can affect the buffer locality & throughput drop.

## Large receive offload

LRO enables the NIC to reassemble received contiguous packets into larger buffers, thus increasing address locality and reducing the number of IOTLB translations. However, the dynamics depend heavily on the buffer size. Figure 6A shows the effect of LRO on the average IOTLB misses per MiB in both underloaded and overloaded conditions. As expected, LRO effectively reduces IOTLB misses per MiB for sub-4-KiB MTUs where the NIC could assemble multiple buffers into a single 4-KiB page. Figure 6B shows that effect of LRO on throughput when operating at overloaded conditions with and without IOMMU; demonstrating that our processor is capable of achieving the same throughput without LRO.

## TX & TCP segmentation offload

This section explores the behavior of the DUT as a sender and studies the impact of IOMMU on the TX path. Figure 7A plots the maximum throughput achieved at different MTUs. Without IOMMU, the sender is close to the maximum theoretical line rate at all MTUs, with or without TSO. Enabling the IOMMU, without TSO, introduces an even larger throughput drop than on the receive side. Figure 7B difference is due to a much

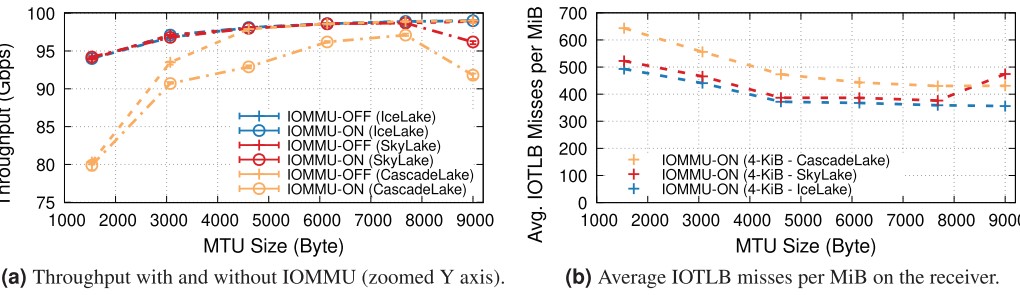

**(a)** Throughput with and without IOMMU (zoomed Y axis).   **(b)** Average IOTLB misses per MiB on the receiver.

**Figure 8 Throughput and IOTLB misses on several Xeon processors, operating at 100 Gbps.**

higher number of IOTLB misses per MiB than that of the receive side (see Fig. 4). Without TSO, each TX packet is made of at least two memory buffers: one for the header, one for the payload, and sometimes more if the payload comes from two different `sendmsg()` calls (as opposed to the receiver where each RX packet uses a single buffer). Larger MTUs reduce the number of segments, resulting in fewer misses per MiB and higher throughput.

With TSO, TX packets sent to the NIC are made of one header followed by up to 64-KiB of data, generally making full use of 4-KiB pages. The NIC takes care of segmentation, thus requesting approximately one translation request per 4-KiB of data (256 per MiB) irrespective of the MTU. This is clearly shown in the TSO-ON curve in Figure 7B, with an approximately constant ∼300 misses/MiB (the extra ones, once again, come from accesses to descriptors, interrupts, and next-level IOMMU entries).

## Different processors

This section compares the impact of IOMMU on different processors. The key difference, for our study, is the PCIe bus version, which in turn affects the maximum PCIe bandwidth, NIC link speed, and also influences the vendor design choice of the maximum number of outstanding PCIe transactions supported. In fact, PCIe "posted data" credits, a hardware feature that affects the maximum number of outstanding PCIe write transactions, tend to be proportional to the bus speed (*i.e.*, PCIe-4 buses targets approximately twice the number as PCIe-3 ones).

We separate the evaluation into two parts: 100 and 200 Gbps, and present throughput and IOTLB misses per MiB both with and without IOMMU.

**100-Gbps link speed.** This set of experiments uses the NIC at 100 Gbps, which is the highest supported NIC link rate on PCIe-3.0. We compare Intel Skylake & CascadeLake (with PCIe-3.0) and our previously used IceLake (with PCIe-4.0). The server equipped with the Intel IceLake processor uses a different NIC than previous experiments. Appendix A describes our testbed.

Figure 8A shows the receiver throughput on the various platforms with and without IOMMU at different MTUs. Noticeably, one of our workstations is unable to achieve line rate at the smaller MTUs[10].

Enabling IOMMU further differentiates the behavior with IceLake still being able to achieve line rate across the board, whereas SkyLake and CascadeLake experience

---

[10] Some tuning might help, but that is beyond the scope of this study.

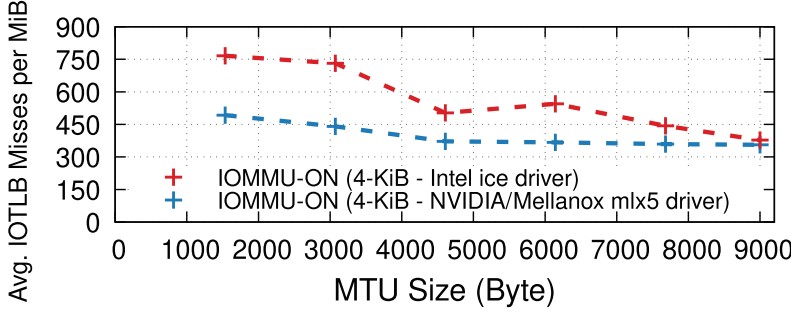

**Figure 10 IOTLB misses of Intel E810 and NVIDIA/Mellanox ConnectX-6 NICs at 100 Gbps.**

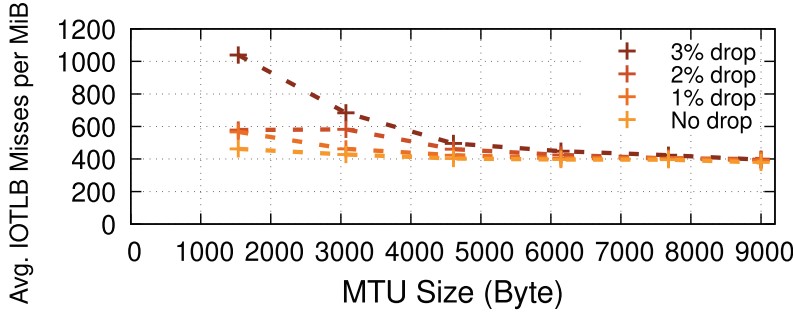

**Figure 11 Packet drops put more pressure on the IOTLB due to buffer shuffling and poor recycling.**

Furthermore, we observed that both drivers often perform continual page allocations at high data rates.

## Other workloads

Our study has mainly focused on iPerf microbenchmarks that represent applications capable of running at line rate with datacenter-like RTTs and limited drops. This section concludes our evaluation by measuring IOTLB misses in two other scenarios considering (*i*) workloads with higher packet drop rates and (*ii*) a less I/O intensive application (*i.e.*, Memcached).

**Packet drops.** Drops (when using TCP) could cause receive buffers to be held longer in the Linux kernel before being passed to the application due to out-of-order delivery, negatively affecting the buffer recycling and increasing shuffling (see "RX path"), thereby causing increased IOTLB misses. To measure the impact of longer receive buffer lifetimes (caused explicitly by packet drops) on the number of IOTLB misses, we developed a kernel patch that artificially drops packets (in the RX path) in the kernel network stack to induce TCP re-transmissions. Figure 11 shows that the higher the drop percentage, the higher the number of IOTLB misses per MiB. In other words, longer buffer lifetimes put greater pressure on the IOTLB. For instance, dropping ∼3% of the packets for every TCP flow could increase IOTLB misses per MiB by up to ∼2.5×.

**Memcached.** To show the applicability of our evaluations to less I/O intensive applications, we measure the number of IOTLB misses per MiB for a Memcached

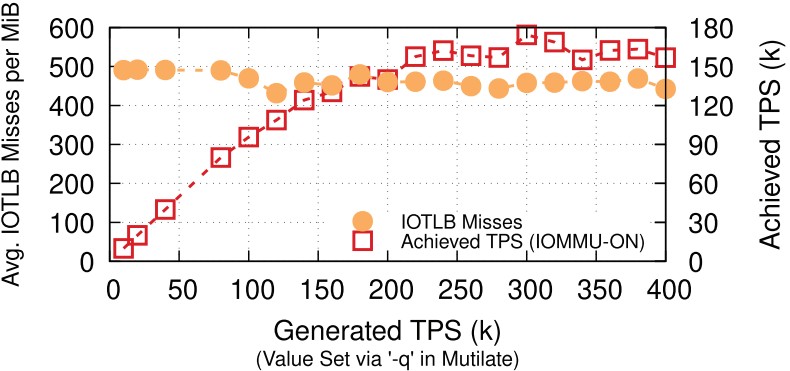

**Figure 12 Memcached (a less I/O insensitive application) has a similar IOTLB pressure to iPerf at 90 Gbps (TPS = ~170k).**

[12] We choose a workload similar to DAMN (*Markuze et al., 2018*). Using the 50/50 SET/GET workload generates bidirectional traffic (as opposed to iPerf), which puts pressure on the IOTLB on both RX and TX.

workload. We use a 32-core Memcached-SR (*Ghigoff et al., 2021*) & Mutilate (*Leverich & Kozyrakis, 2014*) and 64-B keys & 128-KiB values with 50/50 SET/GET workloads[12] to achieve a comparable rate (90 Gbps when TPS = ~170k) to our iPerf measurements. Even though Lancet (*Kogias, Mallon & Bugnion, 2019*) categorizes Mutilate as an open-loop benchmarking tool, developing a more efficient workload generator could result in higher throughput, causing even more IOTLB misses per MiB. Figure 12 shows that Memcached causes 450–500 IOTLB misses per MiB (same range as for iPerf), despite being less I/O intensive and having other throughput bottlenecks. We interpret the existence of these iPerf-comparable results as a validation of our choice to use iPerf as the main driving application (in addition to its inherently deterministic behavior).

## Summary

**Table 2 Takeaways from IOMMU & IOTLB characterization.**

| Scenario | Effect |
| --- | --- |
| Increasing traffic rate | After a certain point, excessive IOTLB misses causes a throughput drop. |
| Increasing MTU size | Results in fewer translations and consequently fewer IOTLB misses & less of a throughput drop. |
| Enabling LRO | Reassembles contiguous sub-4-KiB buffers into a single 4-KiB page, reducing IOTLB misses. |
| Enabling TSO | Transfers larger messages to the NIC and consequently results in fewer translations & IOTLB misses. |
| Different hardware | Using different processors & NICs could significantly change the overheads of IOMMU, due to the implementation of the IOTLB & the NIC driver. |
| Workloads with higher packet drop rate | Packet drops & TCP re-transmissions prolong the lifetime of received buffers, causing buffer shuffling and increasing the number of IOTLB misses. |

This section showed that IOMMU is a major performance bottleneck when the offered network data rate approaches 200 Gbps. Using LRO & TSO can reduce the number of

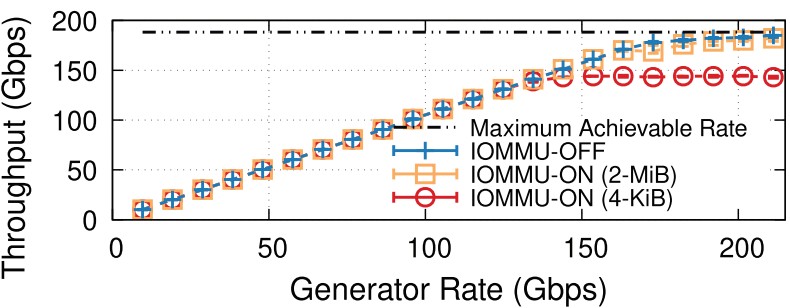

**Figure 13 Throughput comparison with IOMMU-OFF, IOMMU-4-KiB, and IOMMU-2-MiB for MTU = 1,500.** The use of 2-MiB mappings recovers the throughput drop introduced by the IOMMU.

IOTLB translation and partly mitigate the IOTLB overheads, especially for small MTU sizes but still cannot eliminate the overheads at full line rate. Therefore, it is essential to come up with more effective solutions. Table 2 summarizes the main takeaways of our experiments in this section.

## A SOLUTION TO IOTLB WALL

In addition to 4-KiB entries, current processors support IOMMU mappings with 2-MiB and 1-GiB entries. Having established that reducing IOTLB misses, even by a modest amount, has great benefits on throughput (Fig. 3), the use of larger mappings is an obvious way to increase the chance that multiple buffers share the same IOMMU entry.

To prove the effectiveness of this approach, we ran a crude experiment[13] backing buffers with 2-MiB memory "hugepages" and mapping them with 2-MiB IOMMU entries. Figure 13 suggests that with such large mappings, we can achieve almost the same throughput as without IOMMU. This experiment is performed on the same 200-Gbps testbed as Fig. 1, *i.e.*, Intel Xeon IceLake processors, see Appendix A for more information.

Given these preliminary results, our proposed solution focuses on how to build a hugepage-aware buffer allocator to be used for NIC buffers. In the next section, we discuss challenges that arise in designing this subsystem and explore available options to address them.

### Challenges and design options

**Allocation availability and CPU cost.** The kernel allocates memory in 4-KiB units and uses virtual memory to map non-physically-contiguous pages into larger, virtually contiguous segments to satisfy any request from clients.

The kernel runs background compaction tasks to merge free pages into larger, physically contiguous blocks to satisfy *e.g.*, requests from peripherals that do not support scatter/gather I/O or for performance-sensitive systems that need to reduce MMU and IOMMU translation costs. Part of this effort includes remapping pages in use to some other locations to ease compaction. A 2-MiB page, however, requires a significant amount of coalescing (*i.e.*, 512 contiguous 4-KiB-sub-pages), and in a long-running system with significant memory fragmentation, it may be difficult to find a spare 2-MiB page without asking for explicit compaction.

[13] *i.e.*, without much attention to CPU costs and memory usage.

**Memory stranding.** One common option to obtain hugepages is to reserve them at boot-time for a specific use, but this makes the reserved memory unavailable even when the intended client does not need it. Furthermore, a boot-time reservation requires knowledge of the worst-case memory requirements.

For a network device, that is complicated: on the RX side, there is a minimum amount of buffers needed to populate receive queues, but these buffers are passed to the applications' socket buffer and become unavailable for the NIC until applications consume the data. Similarly, on the TX side, buffers (allocated after the `sendmsg()` call and used for TCP) cannot generally be released until congestion control allows transmission and the remote endpoint sends an acknowledgment.

As a minimum, a sender should allow an amount of buffering equal to the bandwidth-delay product. Frontend servers for residential customers are likely to deal with $O$ (10–100 ms) RTT, which at 200 Gbps is 2–3 GiB of memory. A similar minimum amount may be needed on the receive side to buffer out-of-order data. Occasional CPU overloads resulting in delayed reads or systems with a very large number of outstanding TCP connections may cause the total socket buffer requirements to exceed our estimated bandwidth-delay product size.

Depending on the use case, reserving several GiB of memory for network buffers may not be completely out of the question. However, an interesting feature of our problem is that we do not need to eliminate IOTLB misses completely, but simply reduce their impact just enough to prevent them from becoming the primary throughput bottleneck.

**Memory fragmentation.** Once we solve the problem of obtaining hugepages from the kernel, our memory allocator should be able to handle memory fragmentation by itself: transmit and receive buffers only need to handle MTU-sized blocks of data, each with a possibly different lifetime. The buffer managers, mentioned in "RX path", typically receive 4-KiB pages as input and split them internally into a small number (*e.g.*, 2) of smaller chunks to be used as packet buffers. In our case, the input is 2-MiB pages, split into 512..1,024 units. Over time some of these pages may become unavailable (*e.g.*, because they are stuck in socket buffers that are never read from). We call this phenomenon "buffer leakage" and this is something that requires the buffer manager to slowly allocate new buffers.

**Locality.** Using 2-MiB IOMMU mappings does not guarantee locality if we randomly pick buffers from different hugepages. Our allocator should thus track and, in some ways, support returning contiguous buffers to the NIC when it replenishes the receive queues. Both fragmentation and locality can be addressed with solutions that allow fast access to the state of contiguous buffers (such as a bitmap indicating which chunks from a 2-MiB page are available, combined with existing kernel reference counts on the underlying 4-KiB pages).

## Proposed solution

Given the above constraints, we designed and implemented our initial prototype by interposing a hugepage allocator, HPA, between the NIC memory allocator (currently PagePool API[14], which for us is an opaque block) and the kernel memory allocator. Future

[14] Appendix D elaborates on our current implementation.

versions will completely replace the Page Pool API and integrate functionality into the HPA for better locality and performance. The system works as follows:

- When the HPA needs memory, it calls the kernel memory allocator asking for one 2-MiB page, maps it into the IOMMU, and splits it into 4-KiB fragments kept in its internal cache. It also keeps track of the mapping to undo this mapping when the NIC releases its memory.

In future versions, to avoid blocking in the hot path, the HPA will try to issue kernel requests in the background. If a 2-MiB page is not immediately available, it will fall back to regular 4-KiB pages.

- When the NIC device driver needs new buffers, it passes the request to our HPA, which replenishes the cache if needed and responds with 4-KiB pages from its cache. The NIC then uses the IOVA for the page that has been supplied by the HPA instead of creating (and removing when done) a new IOVA.
- In a steady state, the NIC should be able to recycle pages through its own memory allocator, *e.g.*, Page Pool API, and not issue new requests to the HPA.
- Occasionally, pages may not be recycled and are instead returned back to the kernel ("leaked", from the point of view of the NIC; but fully recovered, in terms of memory usage), triggering more requests to the HPA.

The HPA does not waste memory but does keep pages mapped in the IOVA until its pages have been fully released.

### Discussion and future work

Our current solution is constrained by the desire to avoid massive changes to the existing device drivers and is amenable to several optimizations, especially regarding the handling of corner cases. Regardless, it allows us to study in more detail the behavior of the system when using large IOTLB mappings. Open issues and future work include the followings (some already mentioned above):

**Background page allocation, fallback to 4-KiB pages.** We have not yet implemented this feature, which would be useful in systems with low memory availability.

**Locality improvements.** By completely removing the existing page buffer and absorbing its functionality in our allocator, we can do a better job in feeding the NIC with buffers that are closer in terms of the IOMMU address space, following the indications given previously.

**Transmit side support.** Our allocator currently handles only the receive side. Transmit buffers are allocated during a `sendmsg()`, agnostic to the destination of the message. It should be possible to infer, from the file descriptor, whether such data are destined to a NIC using IOMMU, and then allocate the TX buffer from a pool of memory backed by and mapped as a hugepage. Our current implementation is deployable as-is and it is not dependent on the transmit-side solution.

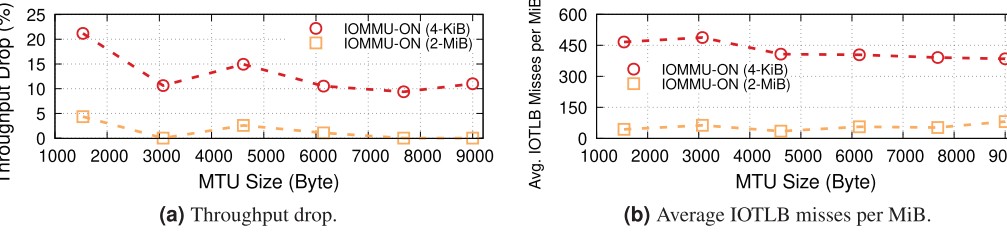

**(a)** Throughput drop.  **(b)** Average IOTLB misses per MiB.

**Figure 14** **Larger IOMMU mappings on the receiver side significantly reduce throughput drop and IOTLB misses.**

**Recovery from thrashing.** Despite our best efforts, it may happen that the system enters a thrashing state where the available buffers cause excessive IOTLB misses. The system should be able to detect this regime (*e.g.*, by looking at NIC packet drop counters on the receive side) and replenish the HPA with a fresh set of hugepages, so that the current, poorly placed pages can be quickly phased out in favor of a new set.

# EVALUATION

This section evaluates the effectiveness of using large IOTLB entries for 200-Gbps networking using our proposed memory allocator. "Frequently Asked Questions" answers some additional questions about our approach.

## IOTLB misses and throughput drop

We already showed in Fig. 13 that the 2-MiB mappings recover almost completely the throughput drop introduced by IOMMU. This is shown in more detail in Fig. 14A, where we see that 2-MiB mappings only causes a modest 5% drop for MTU = 1,500, and enables the system to get very close to the no-IOMMU case for larger MTUs.

Figure 14B compares the average number of IOTLB misses per MiB for the original IOMMU mappings (with 4-KiB IOTLB entries) and our implementation (with 2-MiB IOTLB entries). These results show that employing hugepages helps overcoming the IOTLB wall at 200 Gbps, as the number of IOTLB misses per MiB with 2 MiB entries is reduced (by up to ∼10×, giving us a large safety margin for operation). The gap (350–400 misses/MiB) is in good accordance with our estimates in "Data Rate and IOTLB Wall".

## Run-time allocation overhead

To quantify the overhead of run-time allocation, we benchmarked the execution time of the page-allocation function in the Page Pool API (*i.e.*, `page_pool_alloc_pages_slow`) *via* kstats (*Rizzo, 2020*) over a period of 2 h with an offered load of 200 Gbps. The default buffer allocator requests blocks of 64 × 4-KiB pages to the kernel, whereas our mechanism has a granularity of 512 (contiguous) 4-KiB pages. Individual allocations are thus very likely to take much longer, but their cost is amortized on 8× more pages. The correct model of operation is to make requests for buffers in the background so they do not block critical sections and look at the amortized, per-page cost, which we plot in Fig. 15. The cost distribution for the two scenarios is similar, which is encouraging. We surmise that this

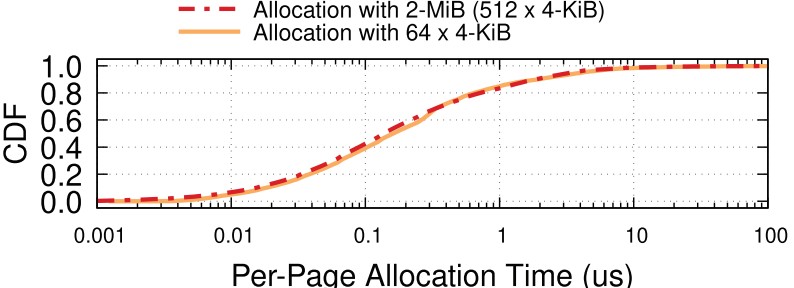

**Figure 15 Per 4-KiB page allocation costs are similar with 4-KiB and 2-MiB pages in our experiments.**

may depend on the relatively low memory pressure in our testbed, making it possible to retrieve 2-MiB pages with relatively little extra CPU overhead.

We should keep in mind that in the presence of high memory pressure, we can always fall back to 4-KiB pages on the grounds that such a scenario will likely introduce other bottlenecks (*e.g.*, CPU or memory contention) that would anyway prevent reaching line rate.

## Buffer shuffling

To conclude, we want to revisit the dynamics of buffer shuffling with hugepage mappings. Our previous experiments with 4-KiB mappings were extremely prone to the problem, which showed up quickly at the first signs of congestion (see "Data Rate and IOTLB Wall"). Due to the nature of the problem, we do not expect it to completely disappear even with hugepage mappings. Additionally, packet drops & TCP re-transmissions can still contribute to buffer shuffling & increase the number of IOTLB misses. To understand the potential overheads associated with buffer shuffling, where the long-term change in the order of buffers causes consecutive pages to have different IOTLB mappings, we ran two types of experiments: ① ("No Pool") where the buffer allocator does not recycle buffers internally, and instead always gets fresh buffers, and ② where the existing allocator recycles buffers very aggressively rather than making new allocations, reusing the same block of 256 or 512 × 2-MiB pages per queue. We only report two block sizes for the recycling case to speculate on the trend. However, the exact behavior per block size might differ in other testbeds. We expect the larger block sizes to have similar behavior with a slower pace/ slope, but the smaller block sizes may be hard to measure, as leaky drivers could deplete a small fixed-size pool quite fast.

Figure 16 shows the IOTLB misses per MiB and throughput over time. All experiments have a modest amount of packet drops. All three cases start with a low number of misses, as expected from our previous experiments. The "No Pool" case stays there for the duration of the experiment, and again this was expected since there is no source of buffer shuffling despite the drops. In the experiment with 512 pages, the number of misses starts increasing after about 2 min, and the curve grows over time as more and more drops cause more severe shuffling (the growth is not unbounded because eventually, there is a hard limit given by the maximum data rate). In the experiment with 256 pages, the shuffling

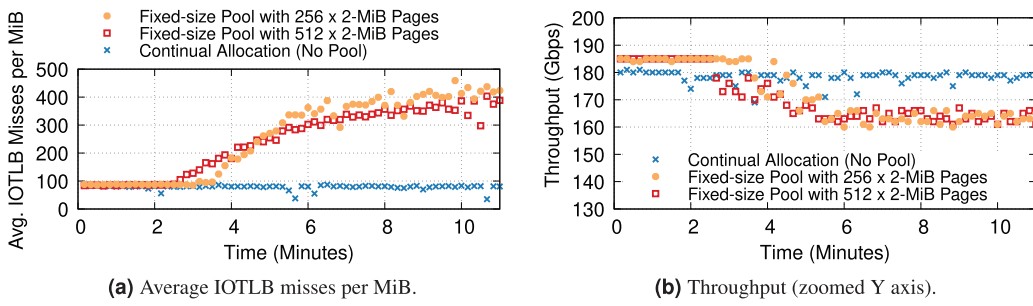

**(a)** Average IOTLB misses per MiB.

**(b)** Throughput (zoomed Y axis).

**Figure 16** Increased IOTLB misses due to the buffer shuffling (shown every 10 s) could cause IOTLB to again become a bottleneck for an iPerf server receiving 1500-B frames with different pool sizes per queue.

appears later but grows more rapidly because the smaller pool gives more opportunities for reshuffling.

## FREQUENTLY ASKED QUESTIONS

**What makes your approach more adoptable?** Previous attempts (*Amit, Ben-Yehuda & Yassour (2012)*; *Malka et al., 2015a*; *Malka, Amit & Tsafrir, 2015b*; *Markuze, Morrison & Tsafrir, 2016*; *Markuze et al., 2018*) focused on improving IOMMU performance, but most of them have not been adopted, speculatively due to the need to introduce hardware modifications (*Amit, Ben-Yehuda & Yassour, 2012*; *Malka et al., 2015a*), imposing other overheads (*Markuze, Morrison & Tsafrir, 2016*), and requiring massive kernel changes (*Markuze, Morrison & Tsafrir, 2016*) (including a larger "struct page" that is a central and fragile kernel data structure). We believe our work is much less intrusive, as it is confined to a buffer allocator without introducing dependencies. Our Page Pool extension can be deployed on a per-driver/module basis as needed without global kernel changes; thus, making experimentation and gradual adoption much simpler and less risky.

**Are hugepages the panacea?** We showed that using statically mapped hugepages can mitigate the performance overheads of IOMMU for high-speed networking. However, hugepages are not a panacea, and faster link speeds would require larger & larger hugepages to achieve optimal performance with the same number of IOTLB entries. Therefore, future-generation hardware is expected to introduce changes to the IOMMU architecture, *e.g.*, adapting the IOMMU page table as proposed by *Malka et al. (2015a)* and/or modifying the IOTLB to support more concurrent translations per second.

**Does using hugepages relax security guarantees provided by IOMMU?** Switching to statically mapped hugepages may cause security concerns if we do not use a specific memory pool, as the deallocated pages might be used by the kernel or an application while still being exposed to an I/O device (*Alex et al., 2021*). However, (de)allocating hugepage-backed buffers to/from a specific (I/O) memory pool addresses these concerns. Note that each I/O device is limited to its domain mappings; therefore, using larger IOTLB entries only provides the same I/O device with a larger access domain and a longer access time, which should not raise any critical concern as long as these huge IOTLB mappings are only used for I/O.

**How about zero-copy mechanisms?** There are a few old (*e.g.*, sendfile and splice) and recent (*i.e.*, MSG_ZEROCOPY (*Bruijn & Dumazet, 2017*; *Corbet, 2017*) and MAIO (*Markuze, Golikov & Dar, 2021*)) techniques to pass userspace buffers directly to the kernel to avoid data copying overheads; however, most of these techniques are generally only effective for large messages (*Kesavan, Ricci & Stutsman, 2017*) (*e.g.*, 10 MiB (*Linux Networking Documentation, 2021*)). Our approach is not designed with zero-copy mechanisms in mind. As for the RX path, our approach may work for the recent zero-copy receive mechanism introduced by Eric Dumazet (*Corbet, 2018*; *Dumazet, 2020*) & VMware's I/O memory allocator, called MAIO (*Markuze, 2021*; *Markuze, Golikov & Dar, 2021*). However, it could raise additional security concerns since DMA attacks may affect the user/application workflow. As for the TX path, we rely on the current Linux kernel implementation to map buffers to IOMMU. Drivers usually assume that the allocated buffers (*i.e.*, in the RX path) and/or received skbuffs from the kernel (*i.e.*, in the TX path) are backed by PAGE_SIZE (*i.e.*, 4-KiB) pages. Therefore, they create a 4-KiB IOMMU entry even if a buffer is allocated from a hugepage (*e.g.*, a 2-MiB/1-GiB hugepage) in userspace.

**Why should I care about IOMMU overheads in the Linux kernel rather than simply using hugepages in kernel-bypass frameworks?** While bypassing the kernel provides better performance by avoiding some inherent kernel overheads (*e.g.*, data copying), it excludes applications from benefiting from the available network infrastructure in the Linux kernel; thus, requiring programmers to re-develop & maintain equivalents of them in userspace. In particular, many networking applications require different network stacks (*Zhang et al., 2021*), which has resulted in some userspace network stacks, such as mTCP (*Jeong et al., 2014*), f-stack (*Tencent Cloud, 2021*), and TLDK *FDio (2021)*. However, these userspace stacks are rarely maintained for a long period and may (& do) contain bugs. Consequently, many applications are still being deployed that use the Linux kernel (despite its known overheads); therefore, we believe it is important to be able to mitigate IOTLB overheads in the Linux kernel.

## RELATED WORKS

The most relevant works to our article include: *Peleg et al. (2015)* which introduces deferred IOTLB invalidation and then optimizes the implementation of `dma_map()` and `dma_unmap()` to minimize locks and waiting time to allocate an IOVA. They introduced a cache for recently freed IOVAs to avoid accessing the red-black tree holding pairwise-disjoint ranges of allocated virtual I/O page numbers. *Markuze et al. (2018)* presented a memory allocator, called DAMN, to provide both security and performance. This allocator used permanently mapped buffers for IOMMU to avoid performing extra map/unmap operations. DAMN focuses on managing 4 KiB buffers and requires changes to the page data structure. *Markuze, Morrison & Tsafrir (2016)* proposed a copy-based method to improve IOMMU protection (*i.e.*, by solving deferred IOTLB invalidation and sub-page vulnerability). Their method uses a set of permanently mapped buffers called "Shadow Buffers" to perform DMA and then copy the data into applications' buffers. They show that using shadow DMA buffers can achieve better performance compared to other

IOMMU methods (while solving the two mentioned IOMMU problems). *Malka et al. (2015a)* introduce a flat table to improve the performance of IOMMU. This table is based on the characteristics of circular ring buffers. Other IOMMU-related works can be grouped into three categories: ① investigating the IOMMU security vulnerabilities & the possibility of performing different DMA attacks (*Alex et al., 2021*; *Gui et al., 2021*); ② optimizing the performance of IOMMU for hypervisors and virtualized (*Tian et al., 2020*; *Amit et al., 2011*; *Willmann, Rixner & Cox, 2008*; *Lim & Nieh, 2020*; *Lavrov & Wentzlaff, 2020*) and ③ presenting hardware proposals for future-generation IOMMU technologies (*Hao et al., 2017*; *Lavrov & Wentzlaff, 2020*; *Malka et al., 2015a*). Additionally, *Lesokhin et al. (2017)* enable Infiniband and Ethernet NICs to support I/O page faults. Moreover, *Agarwal et al. (2022)* measure the impact of IOMMU on host interconnect congestion. Our article is complementary to these works.

## CONCLUSION

Ideally, networked systems should ensure privacy & security while achieving high performance, but existing systems fail to provide these properties. IOMMU is one of the available technologies that was introduced to ease I/O virtualization and to provide security & isolation. However, high-performance systems tend to disable IOMMU to mitigate its performance overheads, thus voiding the security & isolation guarantees. This article empirically studies IOMMU in recent hardware and demonstrates that IOTLB overheads become even more costly when moving toward 200-Gbps networking. It is possible to alleviate these overheads in software by employing hugepage IOTLB mappings, but there are many challenges in permanently eliminating these overheads. Our main takeaway is that supporting the upcoming 200/400-Gbps networking demands fundamental changes in Linux kernel I/O management.

## A TESTBED ILLUSTRATION AND ADDITIONAL RESULTS

Figure 17 illustrates our testbed. We use the main slot of a Socket Direct ConnectX-6 for 200-Gbps experiments and a normal ConnectX-6 NIC with custom Dell firmware for 100-Gbps experiments.

Figure 18 shows the throughput for a fine-grained sweep of MTU values when the NIC uses 1,024 RX descriptors per queue (*i.e.*, 32 × 1,024 RX descriptors). Looking at the throughput values, we notice jumps at page splitting points. The `mlx5` driver splits pages based on the Hardware (H/W) MTU size that is 22 B larger than the value specified by the software (*e.g.*, *via* `ifconfig`). The extra 22 B ensures that the H/W buffer has enough space for the 14-B Ethernet header, 4-B VLAN header, and 4-B Ethernet Frame Check Sequence (FCS). The `mlx5` driver uses 512-B buffers for small H/W MTUs, but switches to 1,024-B, 2,048-B, and 4,096-B buffers when the H/W MTU size exceeds 128, 640, and 1,664 bytes. The driver uses a scatter-gather technique (*e.g.*, it uses multiple 256-B buffers) for "Jumbo" frames, *i.e.*, H/W MTU sizes > 3,712 B.

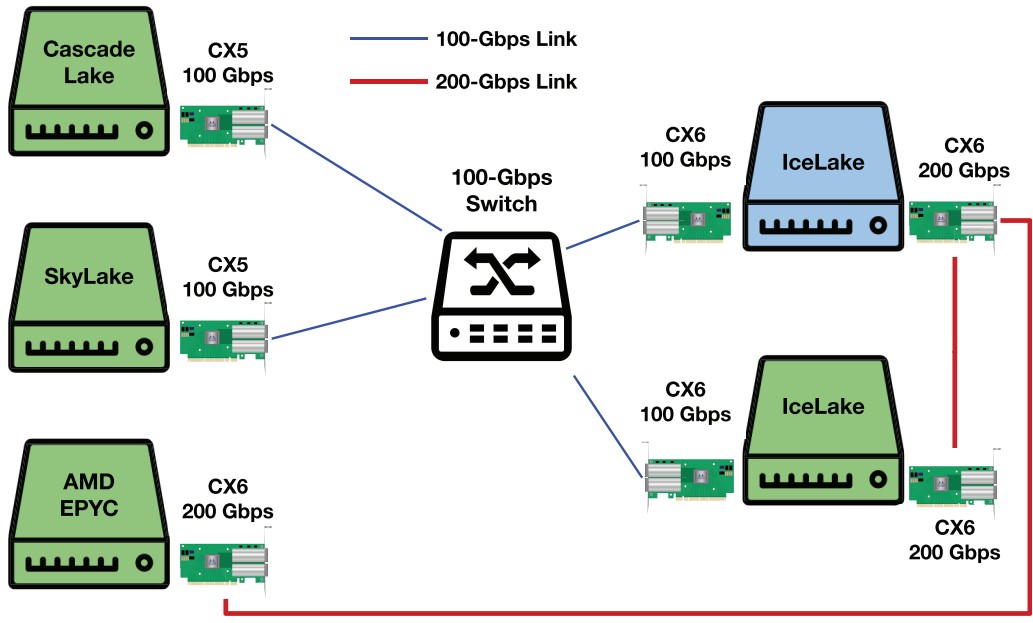

**Figure 17 Our testbed.** Green servers represent DUTs. CX5 and CX6 stand for ConnectX-5 and ConnectX-6, respectively.

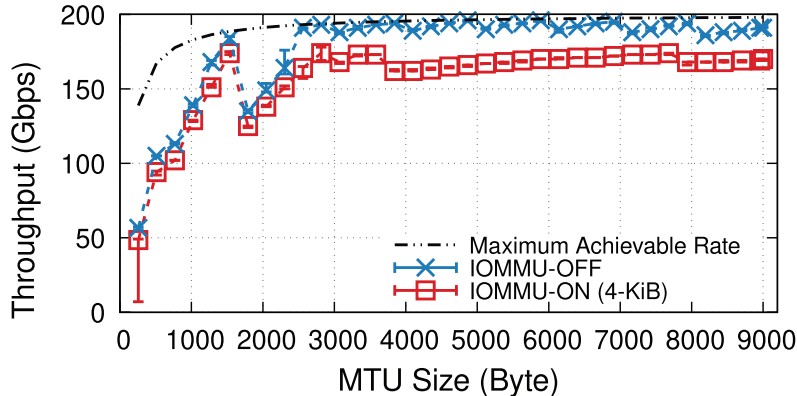

**Figure 18 IOMMU imposes performance overheads at all MTU sizes, but with a larger absolute throughput drop for MTU sizes larger than ∼3,000 B.**

## B IOTLB OVERHEADS IN DPDK

For the sake of completeness, this section examines a kernel-bypass framework, Data Plane Development Kit (DPDK), to examine IOMMU in a high-performance & low-overhead setting. Additionally, using DPDK provides us with greater flexibility in memory management, enabling us to natively compare the impact of different (huge) page sizes (*i.e.*, 4-KiB & 2-MiB) on the performance of IOTLB. Furthermore, since DPDK allocates buffers contiguously (especially when we reserve hugepages at boot time), changing the (huge)page size will primarily show the impact of IOTLB misses. We use a DPDK-based packet-processing framework, FastClick (*Barbette, Soldani & Mathy, 2015*), to generate & forward fix-sized packets at line rate. By doing so, we can benefit from all software

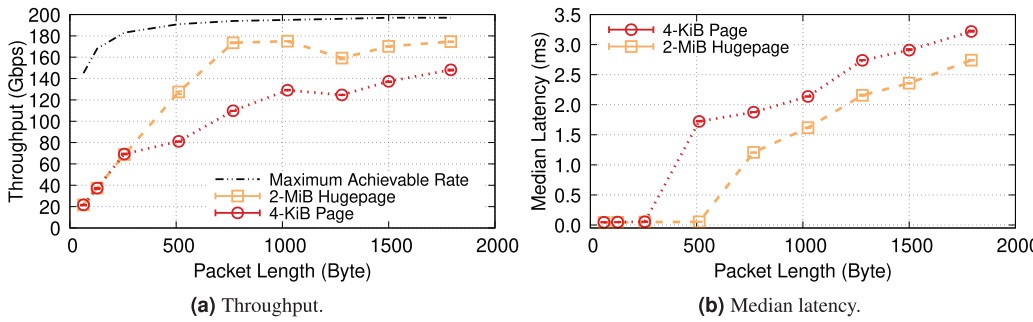

**Figure 19** DPDK also experiences performance degradation with IOMMU when it uses 4-KiB pages.

optimizations shipped within FastClick and achieve the maximum achievable rate with
DPDK (*Farshin et al., 2021*).

Figure 19 shows the throughput and median latency of a server running an L2 forwarder
application that receives fixed-size packets with 16 cores and 1,024 RX descriptors per core,
mirrors the MAC address of received packets and sends them back to the packet generator.
These results show that kernel-bypass frameworks also experience similar performance
degradations when they do not use hugepage IOTLB mappings. Note that the exact
numbers may not match the iPerf results, as DPDK is less CPU-bound (*i.e.*, it achieves
higher throughput with hugepages) and is less efficient without hugepages (*i.e.*, it achieves
lower throughput in 4-KiB mode (*Yao & Hu, 2018*)).

## C PAGE (DE)ALLOCATION *VIA* PAGE POOL API

Page Pool provides an API to device drivers to (de)allocate pages in an efficient, and
preferably lockless way (see Fig. 20 for details in Linux kernel v5.15); the driver then splits
each page into single or multiple buffers based on a given MTU size, see "Effect of MTU
Size". A page pool is composed of two main data structures: (*i*) a fixed-size lockless array/
Last In, First Out (LIFO) (*aka* cache) and (*ii*) a variable-sized ring implemented *via* a
`ptr_ring` data structure that is essentially a limited-size First in, First Out (FIFO) with
spinlocks and that facilitates synchronization by using separate locks for consumers &
producers. By default, the cache can contain up to 128 pages, and the size of the ring is
determined based on the number of RX descriptors and MTU size. The purpose of the
page pool is to (*i*) efficiently allocate pages from the cache without locking and (*ii*) use the
ring to recycle returned pages. To avoid synchronization problems, each page pool should
be connected to only *one* RX queue, thus it is protected by NAPI scheduling. The Page Pool
API is mainly used to allocate memory at the granularity of a page; however, recent Linux
kernels support page fragments that make it possible to allocate an arbitrary-sized memory
chunk from an order-n page, *i.e.*, $2^n$ contiguous 4-KiB pages.

**Details of the steps in Fig. 20**

① Allocating a page. When a driver asks for a page: (1.1) the page pool initially checks
the lockless cache; (1.2) if empty, it tries to refill the cache from the pages recycled in the
ring; (1.3) if not possible due to software interrupts (softirq) or unavailability, it allocates

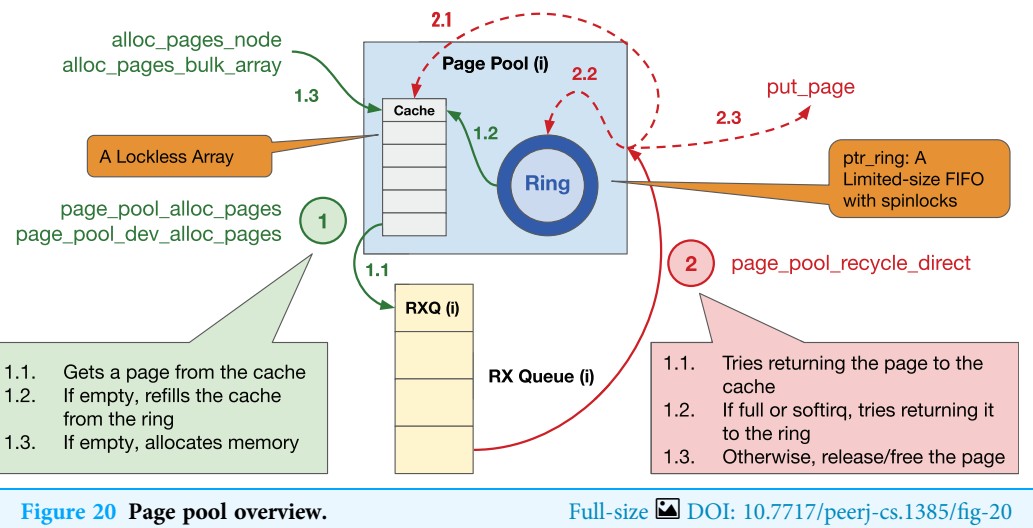

**Figure 20  Page pool overview.**

page(s) and refills the cache. Recent Linux kernels perform bulk allocation (*e.g.*, 64 pages at a time) to amortize the allocation cost.

② Returning a page. When a driver returns a page: (2.1) the page pool attempts to recycle the page into the cache; (2.2) if not possible, it tries returning it to the ring; (2.3) if unsuccessful, it releases/frees the page. Since the Page Pool API is optimized for eXpress Data Path (XDP) and AF_XDP socket (XSK), *i.e.*, an eBPF data path where each page is only used by one buffer, Steps 2.1 & 2.2 can only be performed if (*i*) a page has only a single reference (*i.e.*, `page_ref_count(page) == 1`) and (*ii*) a page is not allocated from the emergency pfmemalloc reserves. The former causes a page pool to continually allocate new pages for drivers operating in a non-XDP mode that splits each page into multiple fragments and uses page references for bookkeeping/recycling. Unfortunately, continuous page allocation disables the driver from re-using a fixed set of pages, causing low page locality. We refer to this as the leaky page pool problem.

# D CURRENT IMPLEMENTATION OF HPA

The current implementation of HPA extends the Page Pool API (*The kernel development community, 2021*) with a backup ring (`ptr_ring`) data structure to store a set of hugepage-backed 4-KiB pages; we chose the `ptr_ring` data structure due to its synchronization efficiencies.

We (*i*) allocate 2-MiB hugepages (*via* the buddy page allocator[15]), (*ii*) map them to IOMMU, (*iii*) keep the mappings in a doubly linked-list to be able to unmap the pages at the page pool's destruction, and (*iv*) store the sub-4-KiB pages in the backup ring. While it is possible to store the pre-allocated memory in the existing cache or ring data structures, we consciously decided not to do so to avoid affecting the existing machinery. In our approach, the backup ring is only used to refill the cache when the page pool needs to allocate memory (see step 1.3 in Fig. 20). If the driver is not leaky, then the backup ring would never be empty (*i.e.*, the pre-allocated memory is sufficient). However, drivers with a leaky page pool will eventually deplete the backup ring. We noticed that our NIC could deplete a 1-GiB per-queue memory (*i.e.*, 512 × 512 pages) after ∼100 s when operating at

---

[15] It is currently limited to allocating order-11 (8-MiB) pages.

200 Gbps. Unfortunately, constantly refilling the backup ring with pages could become costly, as it requires using spinlocks. Therefore, when the backup ring is known to be depleted, we switch to an alternative mode where we allocate a new 2-MiB hugepage and directly inject its $512 \times 4$-KiB pages into the cache data structure. To do so, we increase the maximum size of the cache data structure to 1,024 to potentially reduce the frequency of run-time allocations. Note that the page pool returns the *unrecyclable* pages (*i.e.*, individual 4-KiB pages within 2-MiB hugepage) back to the kernel and only saves their IOVA mappings. Therefore, run-time allocations do not over-allocate memory, but they may fragment memory over a long time and thus make the application more CPU-bound. However, as long as the application falls into the IOTLB-bound region, our solution would improve its performance.

## ACKNOWLEDGEMENTS

We would like to thank Gerald Q. Maguire Jr. for providing insightful feedback on the article and proofreading the manuscript.

### Funding

This work has been supported by an unrestricted gift received for Google Ph.D. Fellowship 2021 in Systems and Networking. The work was also funded by the Swedish Foundation for Strategic Research (SSF). This project has received funding from the European Research Council (ERC) under the European Union's Horizon 2020 research and innovation programme (grant agreement No 770889). The funders had no role in study design, data collection and analysis, decision to publish, or preparation of the manuscript.

### Grant Disclosures

The following grant information was disclosed by the authors:
Google Ph.D. Fellowship 2021 in Systems and Networking.
Swedish Foundation for Strategic Research (SSF).
European Research Council (ERC) under the European Union's Horizon 2020: 770889.

### Competing Interests

Luigi Rizzo & Khaled Elmeleegy are employees of Google.

### Author Contributions

- Alireza Farshin conceived and designed the experiments, performed the experiments, analyzed the data, performed the computation work, prepared figures and/or tables, authored or reviewed drafts of the article, and approved the final draft.
- Luigi Rizzo conceived and designed the experiments, analyzed the data, authored or reviewed drafts of the article, and approved the final draft.
- Khaled Elmeleegy analyzed the data, authored or reviewed drafts of the article, and approved the final draft.

- Dejan Kostić analyzed the data, authored or reviewed drafts of the article, and approved the final draft.

## Data Availability

The code is available at Zenodo:

(1) General description and experiments: Alireza Farshin. (2022). aliireza/iommu-bench: PeerJ-CS (v1.0). Zenodo. https://doi.org/10.5281/zenodo.7278161.

(2) A kernel branch to allocate hugepage-backed buffers within PagePool API: Alireza Farshin. (2022). aliireza/linux: PeerJ-CS-hugepage (v5.15-hugepage). Zenodo. https://doi.org/10.5281/zenodo.7278172.

(3) A kernel branch to perform artificial packet drops: Alireza Farshin. (2022). aliireza/linux: PeerJ-CS-Drop (v5.15-drop). Zenodo. https://doi.org/10.5281/zenodo.7278174.

The raw data of our experiments is available at Zenodo: Alireza Farshin. (2023). Raw Data for iommu-bench: PeerJ-CS [Data set]. Zenodo. https://doi.org/10.5281/zenodo.7727193.

## Supplemental Information

Supplemental information for this article can be found online at http://dx.doi.org/10.7717/peerj-cs.1385#supplemental-information.

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
