# Peer review of "Overcoming the IOTLB wall for multi-100-Gbps Linux-based networking"

_PeerJ Computer Science, doi:10.7717/peerj-cs.1385_

## Round 0.1 · original submission · Major Revisions

The authors should address review comments to revise the manuscript.

·

Basic reporting

no commend

Experimental design

no comment

Validity of the findings

no comment

Additional comments

Dear authors,

thank you very much for submitting your work to PeerJ. I enjoyed reading your paper and I found your elaborate analysis and findings very useful. Thank you for going through all these experiments. I believe that the systems and networking communities will appreciate the contribution. I personally valued section 3 much more than S4 that describes your mitigation approach.

I only have a few minor suggestions and comments to improve the way you introduce your contribution:

* I found the IOTLB misses per MiB metric very interesting and exactly the one you want to use for your experiments. I think you should explain this metric more in detail though, since it is used extensively in your paper. Does it refers to the transmitted or received data? What happens when there are packet retransmissions, does MiB increase? Do you consider only application-level data, i.e. goodput, or you count headers in that?

* In 3.7 you introduce your methodology to evaluate the impact of packet drops based on the kernel patch. What does this patch impact? The TX or the RX path, i.e. do you drop packets on the sender or the receiver?

* I would appreciate one extra sentence describing why lost packets lead to longer buffer lifetimes on the receiver end. The out-of-order delivery was only implied and it wasn’t clear. My understanding is that this only becomes a problem when there are following packets for the same connection that are not dropped. Initially, I thought you referred to the sender side, since now the dropped packet needs to be retransmitted and stay around for longer.

* It wasn't clear to me why you introduced the memcached experiment in Section 3.7. I think the important aspect in this experiment that you didn’t evaluate before is the impact of the packet size. In your Memcached experiment you have small less than an MTU sized packets. I assume this is why the IOTLB misses per MiB are slightly higher compared to your previous experiments. I think that having such an experiment before, that shows the IOTLB misses for smaller packets will be more interesting.

* In the same Mecached experiment, I found the x axis in Figure 12 to be misleading. Mutlilate is a closed loop load generator. So, it cannot consistently generate more requests than what the server can actually server. I can see why going beyond the server capacity might make sense, but for that you will need an open-loop generator.

* I couldn’t understand the difference between Figure 13 and Figure 14 for the case of MTU=1500. Is the 5% performance drop still present but not visible in figure 13?

* For the packet shuffling problem and evaluation, you could include another figure similar to Figure 11 that shows the impact of packet drops when 2MB pages are used.

* Unlike other system papers that require some evaluation for the motivation of their design and the major part of the evaluation focuses on the proposed design, the extensive evaluation for motivating the solution is the main contribution of this paper. Thus, I would suggest you focus on the analysis more and only introduce the solution as a potential direction and work in progress. Otherwise, you should go through a robust evaluation of the proposed solution in which you'll have to compare with different baselines, evaluate the allocation time (as in Fig15) under different workloads and memory pressure, etc. Instead, you can run the same set of benchmarks that you run in Section 4 with a statically allocated pool of 2MB pages, even if it is wasteful. This will show the maximum performance benefits and will complete the analysis. For example, I'd be interested to see what is the impact of packet drops when 2MB mappings are used?

* Based on the previous comment, I can also see a version of the paper that does not include the hugepage-aware buffer allocator as a contribution at all and only hints it as a potential design. To me, the complete experimental analysis, with 4kb and 2MB sized mappings, alone is a great contribution for a journal paper.

·

Basic reporting

Figure 6a does not mention the label on the "x-axis" ---> I suppose it should be "MTU size (Byte)

Experimental design

The paper presents a well-detailed evaluation of IOMMU impact on the network throughput of a Linux-based server.
The paper acknowledges previous studies and pushes the evaluation in the context of multi-gigabytes network cards (up to 200GB) in the paper.

I also like the breakdown of the IOMMU impact regarding several parameters, LRO, TSO, and processor type. The results shed light on several potential improvement areas of the network stack that are worth exploring.

However, the authors do not mention the packet sizes used for iperf. It hinders reproducibility and raises the question of the impact of packet sizes (e.g., jumbo frames) on the results presented (even if we have an idea about the Memcached experiment).

Validity of the findings

The papers' main observations show that there is a substantial effort that the research community should make to improve the management of IOMMU for devices as devices' speed will continue to grow to meet the stringent market demands.

The contribution regarding HPA is quite straightforward but simple and clear. It works like a slab allocator tailored for IOMMU in the context of network devices.

However, Figure 14 shows no throughput drop with an MTU size of 9000 with HPA. I find this result surprising. Does this mean that the impact of IOMMU is nullified with HPA for big MTU sizes? Unfortunately, no explanation regarding this surprising observation is provided.

The average IOTLB miss per MiB is hard to reason with. It would be interesting and fairer to provide a table showing the raw number of IOTLB misses during the experiments.

Additional comments

I suggest the authors add information about which packet sizes are used for iperf. It is only mentioned that it is used for its efficiency to saturate a 200GBps link. While observing the results, I asked myself if the packet sizes could also impact how the LRO and TSO behave (especially for JUMBO frames). Providing this information will improve the reproducibility of your results.

My only observation is that the average IOTLB miss per MiB is hard to reason with. Therefore, I suggest augmenting the graphs with a table showing the raw number IOTLB misses during the experiments.

I suggest the reviewer explain why the throughput drop is 0 for MTU size 9000 with HPA (Figure 14). Does it mean that IOMMU does not impact the throughput for bigger MTU sizes with HPA?

I understand the choice of focusing on throughput. Still, it is fair also to show the memory overhead of every scenario to provide more information to readers to reason with the whole picture. Moreover, it will provide a bigger picture of what is happening for someone who seeks to propose a solution.

Overall, the study is interesting and shows that as network device capacities continue to grow, kernel subsystems should be redesigned. This raises a question, however, for each capacity leap, should the kernel always adapt itself, or isn't there a way to provide IOMMU with minimal steady memory overhead independently of the capacities of the attached devices?

Reviewer 3 ·

Basic reporting

This paper is written at a level of English which is sufficient for the reader to understand the main claims and not be misled by syntactical errors or ambiguous phrases. That being said, I would recommend that the authors perform a detailed edit of the paper in order to correct the many small grammatical errors it contains. For example, in the first paragraph of the introduction we find: “Most studies focused on…” whereas the text should read “Most studies focus” on, because of agreement. The fluency of the text is better in the first third and decreases towards the end. Despite these criticisms, I would like to explicitly repeat that these small issues were not significant roadblocks to comprehension, and I did not hold such errors against the paper in my review.

The paper’s content follows a fairly standard structure, with two departures. First, the background section is extensive and explains technical content in detail (e.g., memory management for DMAs). Secondly and more significantly, the related work section is in my opinion unacceptably short. I am sympathetic to the authors because much of the literature review is done in the background section, and an extended Related Work would have likely been tautological. However, I do believe that the paper’s structure currently fragments the discussion of Related Work across Sections 1, 2, 6, and 7, which leads to much less clarity in comprehension. I would suggest the authors modify the text to merge Sections 2, 6, and 7, specifically introduce headings to categorize previous works (e.g., CPU overheads of IOMMU, zero-copy networking), and describe how this work adds to the literature.

Finally, the figures are of appropriate quality, the paper is indeed self-contained, and the code is publicly available. In contrast, no raw data is provided with the paper. There are a few small issues with certain figures, which are listed below:
- Figure 6a is missing the y-axis label.
- Figures 13 and 14 are missing any explanation of the workload and system used to generate them. Before describing these figures, the text has used multiple systems and workloads to study different IOMMU behaviours, and therefore it is not clear which setup these experiments use.

Experimental design

I commend the authors for their number of experimental results and the multiple platforms that they investigated during their study. Also, the choice of area is relevant and meaningful, particularly in datacenter deployments where such high-bandwidth NICs and IOMMU usage are very likely to overlap due to resource virtualization and the associated security concerns. Additionally, the authors have clearly gone to a large amount of effort to conduct experiments with state-of-the-art systems and tools, which in my judgment are mostly methodologically sound.

My main criticism of the experimental directions in this paper are that the links between many of the studies and prior work is not clear. For example, the authors write that prior work has focused mainly on the CPU overheads of IOMMU usage (Section 1, lines 47-55 in the submitted PDF), and that their work considers hardware-dependent factors linked to “the time which IOTLB misses take to resolve” (lines 201-206). The paper then goes on to state that it “[characterizes] IOTLB behavior in a variety of scenarios”, (they study MTU, LRO, and TSO to name three), and show broadly similar results between all of them. One may ask, why choose these and not others? The paper’s results would have been much stronger if it first clarified which of the IOMMU’s underlying features is the most relevant for system performance, and based its experiments around those features. For commentary on the similarity between results, see paragraph 1 of the section “Validity of Findings”.

Furthermore, there is a very important related work which would help guide the questions being asked in this paper, which was not considered at all other than a small mention in the Related Works section. Lavrov et al. (HyperTRIO, ISCA’20) performed a detailed study of IOMMU behaviour with the same iPerf workload used by this paper, and showed specifically that I/O bandwidth can become a bottleneck due to a lack of scalability in the IOMMU subsystem. Although HyperTRIO studies virtualized environments, I do not expect their conclusions (e.g., their breakdown of access patterns of applications, TLB associativity, translation working set size, and TLB prefetching) to change. If the conclusions by prior work are incorrect or at least inapplicable here, I would have expected this paper to specifically discuss them or conduct experiments to show this.

Finally, this paper contains a noticeable methodological flaw, in the Memcached experimental parameters and configuration. This paper uses an unrealistic workload of 64B keys and 128KiB values, where the majority of prior works show that key-value caching workloads use predominantly small items (Atikoglu, SIGMETRICS’12 and Yang, OSDI’20 are two examples out of many). I expect the authors have chosen such large values so that their Memcached implementation is I/O bandwidth bound and not IOPS-bound as most key-value stores are. My concern is that this configuration is not representative of the state of common key-value caching workloads and is an overly contrived data point.

Validity of the findings

My summary of the main findings of the paper is as follows:
1) Since IOTLB space is at a premium, features that reduce the translation working set are highly desirable, and lead to throughput improvements (larger MTUs, and the use of LRO/TSO all have the impact of reducing working sets via larger packet buffers or fewer TCP segments).
2) There exists a correlation between packet drops in the kernel and IOTLB miss rates, which negatively impacts throughput.

The first conclusion is intuitive and fairly evident in retrospect, although the results do quantitatively provide evidence for it. However, I have some concerns about the second conclusion, as well as the manner in which it is presented.

Most importantly, the main hypothesis in this paper about buffer shuffling being a potential reason for increased IOTLB miss rates was interesting, and in my opinion the key new idea in this work. However, Figure 3 should not be claimed as a “proof”, because almost no good hypothesis can be conclusively proven. It would be extremely difficult to test all possible parameters, remove hidden variables, and eliminate the possibility of ever gathering further evidence, especially in only one experimental result. Instead, the findings shown in Figure 3 should be discussed as evidence for the hypothesis. Figure 11 is also an experiment which shows evidence for the correlation between packet drop rates and IOTLB misses.

Regarding the claim about buffer shuffling, I did not understand how the IOTLB miss rates can be worse than the fully linear scan pattern explained in the text accompanying Figures 2/3. If the analysis there is correct (and the IOTLB size is indeed 64 entries), then 100% of the RX buffers should incur misses in the IOTLB for 4KB MTUs. Even in the case of packets being dropped and buffers only being recycled (i.e., handed back to the NIC for incoming DMA), how can the modified access pattern be worse than a miss on every single buffer? If the correlation between packet drops and IOTLB misses is really causation, more analysis and clarity about the access patterns leading to misses would be appreciated.

In Section 3.1, the authors state that 1500B MTUs require ~725 RX buffers (I assumed this is per RX/TX ring following the methodological parameters in Section 3), and it is highly likely that they are contiguously placed, thus leading to 1 IOTLB miss per 4KB page frame. However, the text says that this behaviour is enough to cause ~420 IOTLB misses/MB. By their own simple model, the RX buffers should only lead to 725/2 ~= 365 misses/MB. The remaining ~60 are not discussed. However, the text does mention an additional “80-100 misses/MB coming from misses on the next-level entries and interleaving…” Given that only 1MB of RX buffers are being used here, should the next page-table level not be a 2MB region (4KB*512) which is able to translate all the RX buffers with only one entry (and thus only one miss)? Doing the same basic calculation on the experimental results for MTU = 3690 reveals the same gap between the IOTLB misses expected (288 in the text) and what is measured (~350 from Figure 3). If the simple number of RX buffer entries yields an inaccurate estimate of IOTLB misses, why invite this model in the text?

Figure 18 also raises similar questions about the buffer recycling patterns. For example, the “No Pool” line obeys what we would expect for a solution where every 2MB region will incur a miss, because there is no recycling. However, what is special about the 2 minute threshold that the fixed-size pools now begin to incur more misses? Furthermore, since the total buffering size is smaller for a 256 x 2MB pool, I would have expected it to deplete its pool earlier than the 512 x 2MB system and thus incur misses earlier in time as buffers begun to be reused by the NIC. However the figure seems to show the opposite, that the 512 x 2MB system begins to recycle and incur misses earlier. Why is this? The paper doesn't explain this result at all.

Finally, comparing the results in Figure 8 and Figures 2-3 raise some coherence questions. As stated in the text, all three figures measure Ice Lake; Figures 2-3 use a 200Gbps NIC, where Figure 8 uses 100Gbps. At 100Gbps, the processor seems to attain ~500 misses/1 MB compared to only ~420 when running 200Gbps and 1500B MTUs. Wouldn’t we expect that less bandwidth should result in a smaller translation working set (because the system has a smaller bandwidth-delay product) and thus fewer IOTLB misses? This question was not addressed in the paper.

Additional comments

I found this paper to be an interesting read and low-level study of a large number of software parameters and to see their impacts on the IOMMU. From a technical level, I think the main contribution shown here is the correlation between packet drops and IOTLB miss rates. The solution to use hugepages to reduce the TLB’s working sets is reasonable and the implementation is clearly designed by very knowledgeable authors.

To improve this paper, I would suggest the authors mainly concentrate on clarifying the IOTLB access patterns being generated by the various parameters and connecting them to the miss rates being seen. Although the paper is already at the stage where a reader clearly sees some correlation between MTUs, offloads, packet drops, and miss rates from your experimental results, I recommend more explanation of why those results are obtained (see my comments about Buffer Recycling).

---

## Round 0.2 · accepted · Accept

The reviewer has recommended accepting the submission.

·

Basic reporting

no comment

Experimental design

no comment

Validity of the findings

no comment

Additional comments

Dear authors,

thank you very much for the great work in addressing reviewers' comments and improving the manuscript. I don't have anything more to add